# REPOSITORY-LEVEL PROMPT GENERATION FOR LARGE LANGUAGE MODELS OF CODE

## ABSTRACT

With the success of large language models (LLMs) of code and their use as code assistants (e.g. Codex (Chen et al., 2021) used in GitHub Copilot[1]), techniques for introducing domain-specific knowledge in the prompt design process become important. In this work, we propose a framework called Repo-Level Prompt Generator that learns to generate example-specific prompts using prompt proposals. The prompt proposals take context from the entire repository, thereby incorporating both the structure of the repository and the context from other relevant files (e.g. imports, parent class files). Our technique doesn't require any access to the weights of the LLM, making it applicable in cases where we only have black-box access to the LLM. We conduct experiments on the task of single-line code-autocompletion using code repositories taken from Google Code archives. We demonstrate that an oracle constructed from our prompt proposals gives a remarkably high relative improvement of 36% over Codex, showing the quality of these proposals. Further, we show that when we train a model to predict a prompt proposal, we can achieve significant performance gains over Codex and other baselines.

## 1 INTRODUCTION

Large Language Models (LLMs) have demonstrated remarkable performance in natural language processing tasks (Brown et al., 2020; Chowdhery et al., 2022), text-to-image generation (Ramesh et al., 2022; Rombach et al., 2021), protein-sequencing (Rives et al., 2019) and even as a generalized agent (Reed et al., 2022). As opposed to the *pretrain-finetune* paradigm, *prompting* these LLMs has been found to yield good performance even with few-examples (Liu et al., 2021a). A prompt is an input to the LM such that the desired task can be expressed as predictions generated from the LM. Besides providing a mechanism to control and evaluate a LM, prompts have shown to elicit emergent behaviour as well. Examples of this behavior include GPT-3 (Brown et al., 2020) doing better in tasks it has never seen during training and improved reasoning capabilities with few-shot (Wei et al., 2022) and zero-shot (Kojima et al., 2022) prompts that encourage a chain of thoughts. These factors highlight the importance of designing an effective task-specific prompt [2]. However, currently we have limited understanding of how to do this (Reynolds & McDonell, 2021).

LLMs have also been used for modeling source code with impressive results (Austin et al., 2021; Fried et al., 2022; Xu et al., 2022a). In particular, one of the best performing LLM, Codex (Chen et al., 2021), has been deployed as part of GitHub Copilot [1], a state-of-the-art in-IDE code assistant. Despite the growing popularity of LLMs of code, there is no work that systematically tackles different aspects of prompt generation in relation to source code. One such aspect is that when it comes to code, the relevant context to be put in the prompt can come from not just the current file, but also from outside, such as imports and parent classes. Also, depending on the scenario, the relevant context can be scattered across multiple locations. Since the LLMs have a limited context length available for the prompt, it becomes increasing crucial for our domain-specific understanding to guide the selection of relevant context. Currently, it is not clear how to integrate this domain knowledge of what constitutes a relevant context, into the process of creating prompts. Addressing this question has potential benefits in other domains such as question answering (Liu et al., 2022) and multi-document summarization (Xiao et al., 2022), where domain-specific structured retrieval of context can be useful.

---

[1]https://copilot.github.com/
[2]Platforms such as PromptBase `https://promptbase.com/` allow buying and selling of prompts.

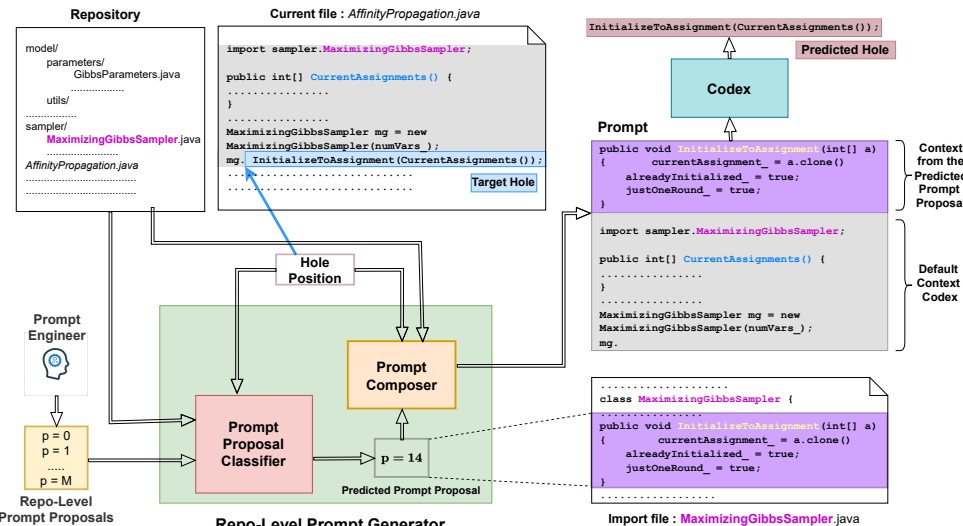

Figure 1: **Repo-Level Prompt Generator:** The prompt is generated by combining the context from the predicted prompt proposal $p = 14$, i.e., method names and bodies from the imported file, MaximizingGibbsSampler.java (violet) with the default Codex context (gray).

In this work, we address this problem by proposing *Repo-Level Prompt Generator (RLPG)*, a framework that while generating the prompt, incorporates both the structure of the repository as well as the relevant context in all the files of the repository. In RLPG, the choice of *where* from and *what to take* from the repository is specified by a set of *prompt proposals*. For example, one of the prompt proposal can be to take all the identifiers used in the first import file. These prompt proposals allow the prompt engineers to induce their domain expertise in the prompt-designing process. With the increasing use of LLMs as assistive agents to humans, demand for transparency and the desire for software engineers to take active part in tailoring prompts to suit their requirements (Jiang et al., 2022; Sun et al., 2022), this capability becomes important. As suggested in some previous works in NLP (Shin et al., 2020; Schick & Schütze, 2021), our prompt proposals are discrete. However, rather than fixing one particular prompt proposal for each example, we instead predict the best prompt proposal conditioned on the example. We do this by coming up with a neural network called *Prompt Proposal Classifier (PPC)*, that given an example, *learns* to select a prompt proposal such that the resulting prompt is likely to produce the desired output. Therefore, RLPG allows the introduction of domain expertise, and at the same time facilitates automatic example-specific prompt generation via a learned neural network. Note that there are some techniques for automatic prompt generation in NLP (Li & Liang, 2021; Shin et al., 2020; Lester et al., 2021) that require updating some or all of the weights of the LLM. However, the strongest LLMs are not publicly available (e.g. OpenAI provides access *only* to the generated output from Codex via an API [3] and no access to model weights and data is provided), making these techniques less useful under this scenario. RLPG addresses this limitation by generating prompts assuming only *black-box access* to the LLM.

We focus on the task of single-line code-autocompletion in an IDE, where the objective is to predict the blanked-out portion (or *target hole*) starting from the position of an imagined cursor to the end of line. We operate under the line-level maintenance setting (Shrivastava et al., 2020; Hellendoorn & Devanbu, 2017) that reflects the scenario where a user is editing an existing file. This means that there can be code following the line. Figure 1 provides an illustration of our approach. The prompt proposal classifier takes in the hole position (position of the cursor) in the current file, the repository to which the current file belongs and a set of repo-level prompt proposals as input, and predicts a prompt proposal. In our illustrated example, the predicted prompt proposal corresponds to taking the method names and bodies from MaximizingGibbsSampler.java (mg. before the hole position indicates that a method from the imported file is likely to be invoked). The *Prompt Composer* uses the context from the predicted prompt proposal and combines it with the *default Codex context*,

---

[3]https://openai.com/blog/openai-codex/

i.e., code prior to the position of the hole in the current file. The resulting prompt consists of the method name `InitializeToAssignment` (from the prompt proposal context) and the method `CurrentAssignments()` (from the default Codex context), resulting in a successful prediction (brown box on the top) of the target hole. Our key contributions are as follows:

- We propose a framework called the *Repo-Level Prompt Generator (RLPG)* that *learns* to generate prompts conditioned on the example, without requiring access to the weights of the LLM.
- To incorporate domain-knowledge in the prompt design process, RLPG uses a set of repository-level prompt proposals. These prompt proposals are designed to incorporate both the structure of the repository as well as the relevant context from all files in the repository.
- On the task of single-line code-autocompletion, we show that an oracle constructed from our proposed prompt proposals gives up to 36% relative improvement over Codex. This improvement is pleasantly surprising as Codex has never seen prompts made from these prompt proposals during training. Further, we show that when we use our prompt proposal classifier to predict the best prompt proposal, we can achieve up to 17% relative improvement over Codex.

## 2 REPO-LEVEL PROMPT GENERATOR (RLPG)

In this section, we provide details of our framework. We start by describing our prompt proposals and then discuss our prompt proposal classifier which is followed by a description of prompt composer.

### 2.1 REPO-LEVEL PROMPT PROPOSALS

The core idea of RLPG consists of substituting part of the default context used by Codex with context coming from somewhere else in the repository. The decision of what to take and from where in the repository to take from is governed by a set of prompt proposals. These prompt proposals were decided based on manual inspection of our training data and intend to capture common coding patterns (but more generally can also include project/organization-specific coding practises). A prompt proposal can be thought of as a function that takes as input a target hole's position and the repository that the hole is a part of, and that returns the *prompt proposal context* (a string constituted by the context from the prompt proposal). A prompt proposal is specified by a prompt source and a prompt context type. We mention each of these along with their motivation below.

**Prompt Source:** For a target hole position, a prompt source determines from *where* should we take code that will be part of the prompt proposal context. We propose ten different prompt sources:

1. **Current:** take code from the current file excluding the contents of the target hole. The current file is the file that contains the target hole. The code in the current file (e.g. the lines after the hole position) can be very useful in predicting the target hole.
2. **Parent Class:** take code from the file that contains the parent of the class to which the target hole belongs. The intuition behind this is to account for cases where a method present in the parent class is invoked in the current file (i.e. the child class).
3. **Import:** take code from the import files used in the current file. The dependencies specified via imports can provide useful cues to predict the target hole.
4. **Sibling:** take code from the files that are in the same directory as the current file. Files in the same directory tend to share code variables (e.g. identifiers).
5. **Similar Name:** take code from files that have a similar name as the current file. Similar names are determined by doing a splitting of the file name based on underscore or camel-case formatting and then matching parts of the filename. If one or more parts matches, two files are considered to have similar names. The intuition behind this is that software developers tend to name files based on the functionality of the code written in that file. Therefore, a similar name file might contain some portion of the code that is common with the current file and hence might be useful for predicting the target hole.
6. **Child Class:** take code from files that have the current file as their parent class file.
7. **Import of Parent Class:** take code from the import files used in the parent class files.
8. **Import of Sibling:** take code from the import files used in the sibling files.
9. **Import of Similar Name:** take code from the import files used in the similar name files.
10. **Import of Child Class:** take code from the import files used in the child class files.

The last four prompt sources are useful when the target hole occurs at the very beginning of the current file. In these cases, there would be less context coming from other prompt sources. For each prompt source, we can get either a single file or a ranked list of files (see Appendix B.1). In the latter case, we will take context from these files until we exhaust the maximum context length allocated to the prompt proposal.

**Prompt Context Type:** The prompt context type determines *what* code to take from the prompt source. We propose seven different prompt context types (Appendix B.2 has examples of each type):

1. **Post Lines (PL):** Take all the lines after the target hole line till we reach the end of the file. This context type is applicable only when prompt source is the current file [4].
2. **Identifiers (I):** Take all the identifiers used in the prompt source.
3. **Type Identifiers (TI):** Take all the type identifiers used in the prompt source.
4. **Field Declarations (FD):** Take all the field declarations used in the prompt source.
5. **String Literals (SL):** Take all the string literals used in the prompt source.
6. **Method Names (MN):** Take all the method names along with their signatures that are used in the prompt source.
7. **Method Names and Bodies (MNB):** Take all the method names along with their signatures and corresponding bodies used in the prompt source.

By combining prompt sources with prompt context types, we get a total of 63 prompt proposals (see Appendix B.4 for details). Note that depending on the target hole, not all prompt proposals would be applicable (e.g. if there are no parent classes in the current file, prompt proposals with prompt source as parent class file won't be applicable). In Figure 1, the predicted prompt proposal corresponds to taking prompt source **Import** and prompt context type **MNB**. We aimed for a set of prompt proposals that offer more diversity rather than a set of prompt proposals that are all good. This in turn ensures that for any hole position, a significant number of prompt proposals are applicable.

## 2.2 PROMPT PROPOSAL CLASSIFIER (PPC)

Given a hole position, the goal of the prompt proposal classifier is to predict the prompt proposal $p$ that will lead to success, where success happens when the predicted hole $\hat{h}$ exactly matches the target hole $h$. This task is formulated as a multi-label binary classification problem since for a given target hole, more than one prompt proposals can lead to success. In this formulation, we treat the default Codex context as one of the prompt proposals. Next, we describe the training procedure for PPC.

**Training:** For each target hole $h$, we generate a ground-truth vector $Y^h = [y_p^h]_{p=1}^M$ which is a multi-hot vector of size $M$, where $M$ is the total number of prompt proposals. This vector is obtained by feeding the prompt generated from prompt proposal $p$ into Codex and then seeing whether $\hat{h} = h$. If there is a match, we say that the prompt proposal $p$ is successful. For hole $h$, if a prompt proposal $p$ is applicable and leads to success, $y_p^h = 1$ and will be zero otherwise. For each hole $h$, we obtain a mask $T^h$ where $T_p^h = 1$ when $p$ is applicable or zero otherwise. The overall training loss $\mathcal{L}$ can be expressed as the sum of individual hole losses $\mathcal{L}^h$ as follows:

$$\mathcal{L} = \frac{1}{N} \sum_{h=1}^{N} \mathcal{L}^h = \frac{1}{N} \sum_{h=1}^{N} \frac{1}{M^h} \sum_{p=1}^{M^h} BCE(\hat{y}_p^h, y_p^h) * T_p^h \quad where \quad M^h = \sum_p T_p^h \tag{1}$$

In the above equation, $N$ is the total number of holes encountered while training, $M^h$ denotes the total number of applicable prompt proposals for $h$ and $BCE$ corresponds to the binary cross entropy loss. Masking ensures that we consider only the prompt proposals that are applicable. Next, we describe our two variants of PPC that can be used to obtain the prediction $\hat{y}_p^h$.

**RLPG-H:** Let $H^h$ be the *hole window* that includes code present around the hole $h$ excluding the hole itself. In our work, we take two lines before the hole position, the code up to the hole position and two lines after the hole position. We use a pretrained model $F_\phi$ to obtain a context representation vector of size $Z$, where $Z$ is the dimension of the hidden state of the model. Specifically, we take the hidden state at the first position, i.e. the representation of the [CLS] token. To make training of PPC

---

[4]We also conducted experiments (Appendix D.2) where we take lines starting from the 4th line after the hole.

computationally efficient, the parameters $\phi$ are frozen during training. The RLPG-H model takes the context representation of the hole window and projects it to the prompt proposal space of size $M$ via two dense layers with a non-linearity in between (see Equation 2). Taking the sigmoid of this output gives the prediction of the prompt proposal.

$$\hat{y}_p^h = P(y_p^h = 1|H^h) = \text{sigmoid}(W^2(\text{relu}(W^1(F_\phi(H^h)) + b^1)) + b^2) \tag{2}$$

**RLPG-R:** The motivation behind this variant is to use the similarity of the hole window and the prompt proposal context to determine which prompt proposal can be useful. Given a particular hole $h$, let $C_p^h$ denote the prompt proposal context from prompt proposal $p$. Intuitively, if the hole window contains variables (e.g. identifiers) that are similar to the variables in the prompt proposal context, then there are chances that $h$ might occur somewhere in $C_p^h$. The similarity is modeled using a multiheaded attention mechanism (Vaswani et al., 2017), by treating the projected hole window representation as a query $Q^h$ and the projected prompt proposal context representation $K_p^h$ as a key (Equation 3). The value $V_p^h$ is the same as the key.

$$Q^h = F_\phi(H^h), \quad K_p^h = F_\phi(C_p^h), \quad V_p^h = F_\phi(C_p^h) \tag{3}$$

$$Att(Q^h, K_p^h, V_p^h) = V_p^h \text{softmax}\left(\frac{Q^{h^\top} K_p^h}{\sqrt{d_k}}\right) \tag{4}$$

$$MultiHead(Q^h, K_p^h, V_p^h) = W^O \text{concat}(head_i, head_2, \dots head_\tau) \tag{5}$$

$$\text{where} \quad head_i = Att(W_i^Q Q^h, W_i^K K_p^h, W_i^V V_p^h)$$

$$\hat{y}_p^h = P(y_p^h = 1|H^h, C_p^h) = \text{sigmoid}\left(W_p G(MultiHead(Q^h, K_p^h, V_p^h)) + b_p\right) \tag{6}$$

In the equations above, $d_k$ is the dimension of the key, $W_i^Q, W_i^K, W_i^V$ are the query, key and value projection matrices, $\tau$ is the number of heads and $W^O$ is the linear projection that combines the heads. The output from Equation 5 is fed to module $G$ consisting of two-layers of feedforward network with relu activation in between (see Appendix C for more details). The resulting output is then linearly projected and a sigmoid is applied to get the predicted prompt proposal (Equation 6).

### 2.3 Prompt Composer

The prompt composer combines the context from the selected prompt proposal (given by PPC) with the context normally used by Codex (default Codex context) to generate the prompt. Since the total length that can be used for a prompt is fixed, we adopted a dynamic context allocation strategy where if the prompt proposal context is shorter than its allocated length, we assign the remaining portion from the prompt proposal context to the default Codex context. The prompt proposal context is always added before the default Codex context. For all prompt proposals, we assign half of the total context length to the prompt proposal context and the remaining to the default Codex context. For post lines, in addition, we also assign one-fourth and three-fourths of the total context length to the prompt proposal context. If the prompt proposal context or the default Codex context is greater than the context length allocated to it, we truncate it (see Appendix B.3 for our truncation strategies).

## 3 Experiments and Results

In this section, we describe our process of dataset creation, details of experiments along with their results and interesting ablation studies.

### 3.1 Dataset Creation

To mitigate the effects caused by potential memorization of the code present in the dataset used for training Codex, we avoided code repositories from GitHub (Chen et al., 2021). Instead, we scraped Google Code [5] for repositories in Java (removing the ones that matched with a repository on GitHub

---

[5]https://code.google.com/archive/

with the same name). We selected the repositories that had a permissive license giving us a total of 47 repositories. We divided the repositories into train, validation and test splits, where each repository in its entirety is part of a split. In each file within a repository, we remove lines that are blank and comments, and set the hole position to be the middle character in the line. All the characters from the middle position to the end of the line constitute the target hole.

Since code duplication has been shown to have adverse effects (Allamanis, 2018), within a repository, we look for files that are exact replica of each other, but placed in a different folder. We mark all such copies as duplicates and omit all of them when creating target holes for our dataset. Note that the prompt proposal context can still come from the duplicate files. We felt comfortable with this choice since we wouldn't want to predict a target hole in a duplicate file, but we can still use the context from the duplicate file to predict the hole in a file that is not its duplicate (e.g. in a sibling file). Further, we found that the repositories were quite uneven in terms of their size. To avoid large repositories dominating the training of PPC, we capped the maximum contribution of holes from a repository to 10,000, i.e. if the total number of holes in the repository exceeded 10,000, we selected 10,000 holes randomly from the total holes.

Please see the left part of Figure 2 for statistics of our dataset. The #Holes represents the holes after deduplication and capping. For some of our prompt proposals, we require semantic information that can be obtained with a parse tree. We used the tree-sitter API for Java [6] that enables us to get the AST of a file and query it. Since our prompt proposals need information at a repository level, we stored some extra information that allowed us to collate the information from individual files according to the directory structure inside the repository (see Appendix A for more details).

## 3.2  EXPERIMENTAL DETAILS

**Prompt Generation:** We used the OpenAI Codex Completions API for generating the predicted hole from the Codex model. In particular, we used the `code-davinci-001` engine with temperature set to 1.0 and stop criteria as newline. The completion length was kept to be 24 and the maximum prompt length was 4072. Tokenization was done using the suggested tokenizer [7]. To allow for fast computation, we used simple models like CodeBERT (Feng et al., 2020) and GraphCodeBERT (Guo et al., 2020) as our pretrained models. One of the limitations of these pretrained models is that the maximum context length that can be taken as input by these models is much smaller than the maximum context length allowed by Codex. Therefore, when getting the representation of the prompt proposal context that is used by PPC, we need to truncate the prompt proposal context that might lead to omitting important parts of the prompt proposal context in certain cases. Using pretrained models that allow larger context length or models that augment the context (Wu et al., 2022) offer avenues for future work. See Appendix D.4 for results when using a smaller context length from Codex.

**Computational Complexity and Scalability of RLPG:** To collect the ground-truth data for training our prompt proposal classifier, we queried the Codex API for each applicable prompt proposal per hole (maximum rate limit of 400 holes per minute). The computational complexity of training our larger RLPG-R variant (3.6M parameters, 141269 holes and 9.19 minutes per epoch on a single Tesla V100 GPU) is much smaller than finetuning all or some part of Codex (12B parameters). During inference, we need to calculate the repo-level statistics just once and all the subsequent hole completions in the repo can utilize this cached information, incurring no additional computational complexity. Besides training the PPC, all our experiments were performed on a CPU with 8GB RAM. Our prompt proposals are based on concepts such as post lines, imports, similar name files, method names and identifiers that are quite general and are applicable to other programming languages. In addition to the existing prompt proposals, our framework provides the flexibility to incorporate new prompt proposals. Since the cost of retraining RLPG with the extended prompt proposals is extremely low (much lower than finetuning Codex with the new prompt proposals), our framework can be used to make interventions on the LLM to address observed weaknesses as long as the intervention can be expressed as a prompt proposal that adds the missing context to the LLM. As opposed to techniques that perform prompt engineering in the latent space and require access to the weights of the LLM such as Li & Liang (2021), RLPG facilitates expressing intent in the form of prompt proposals that are intuitive for humans, easy to understand and do not require access to the weights of the LLM.

---

[6]https://github.com/tree-sitter/tree-sitter-java
[7]https://huggingface.co/docs/transformers/model_doc/gpt2#transformers.GPT2TokenizerFast

| Feature | Train | Val | Test | Total |
|---|---|---|---|---|
| **# Repositories** | 19 | 14 | 14 | 47 |
| **# Files** | 2655 | 1060 | 1308 | 4757 |
| **# Holes** | 92721 | 48548 | 48288 | 189557 |

| Data Split | SR Codex(%) | SR Oracle(%) | Rel. ↑ over Codex(%) |
|---|---|---|---|
| **Train** | 59.78 | 80.29 | 34.31 |
| **Val** | 62.10 | 79.05 | 27.28 |
| **Test** | 58.73 | 79.63 | **35.58** |

Figure 2: *(Left)* Statistics of our dataset; *(Right)* Performance of the oracle relative to Codex.

**Methods:** We experimented with the following methods for generating the prompt:

1. **Codex:** Using the default context from Codex as the entire prompt.
2. **Oracle:** Using the ground-truth vector $Y^h$ (mentioned in Section 2.2). The prompt generated corresponds to using any of the successful prompt proposals (i.e., $y_p^h = 1$). Since this information is not available at inference, the oracle performance represents an upper bound.
3. **Fixed Prompt Proposal:** Using the most successful prompt proposal for all target holes. This was chosen based on the performance on the validation set and corresponded to taking 75% of the total context length from post lines in the current file.
4. **RLPG-H and RLPG-R**: Using the prompt proposal predicted by the RLPG-H and RLPG-H varients of PPC. The selected prompt proposal corresponds to taking the argmax of the predicted probabilities over different prompt proposals.
5. **RLPG-BM25:** Instead of using PPC to rank prompt proposals, using the scores obtained by BM25 (Jones et al., 2000) to select the best prompt proposal. The scores are calculated with the hole window being the query and prompt proposal contexts being the search documents. This serves as a non-learned retrieval method that makes use of our prompt proposals.
6. **File-level BM25:** Same as above, except that instead of using our prompt proposal contexts, search documents consist of full context from other files in the repository.
7. **Random:** For each target hole, select a context randomly from anywhere in the repository.
8. **Random NN:** Same as **Random**, except that amongst the randomly chosen contexts, we take the nearest neighbours of the hole window in the representation space of a pretrained model. This is analogous to the technique used in Liu et al. (2022).
9. **Identifier Usage:** For each target hole, we take the closest identifier and take usage windows of that identifier from everywhere in the repository. We take two lines above, two lines below and the usage line as the usage window. We can rank the usage windows either randomly (**random**) or based on the nearest neighbour distance to the hole window in the representation space (**NN**).

The last four methods help us understand the performance when a context other than the prompt proposal context is used. To generate a prompt using these methods, we take 50% of the context from these followed by the default Codex context that takes up the remaining context length. For the NN baselines, we use CodeBERT (Feng et al., 2020) as the pretrained model. The contexts are taken in the increasing order of the nearest neighbour distances, until we exhaust the allocated context length. RLPG-BM25 helps us understand the role of PPC. See Appendix C.3 for more details on the implementation of these methods.

**Evaluation Metric:** As mentioned in Section 2.2, to measure success, we used exact match between the predicted hole string generated by Codex and the target hole string. In our experiments, we report the percentage of successful holes divided by the total number of holes for each split. We will call this *success rate* (SR) going forward.

### 3.3 RESULTS

In this section, we present the results of the following two research questions explored in this paper:

**[RQ1]** Is it useful to generate a prompt that is composed of code context that is different from the default Codex context? If yes, what context can be useful?

**[RQ2]** For each target hole, is there a way of automatically selecting the prompt? If yes, how does this system perform relative to Codex?

**RQ1 - Performance of Prompt Proposals:** We found that combining the prompt proposal context (context from other files in the repository) with the default Codex context led to substantial improve-

Table 1: Success Rate (SR) of different methods on the test data when averaged across all holes (hole-wise) and across individual repositories (repo-wise)

| Method | Success Rate(%) (hole-wise) | Rel. ↑(%) (hole-wise) | Success Rate(%) (repo-wise) | Rel. ↑(%) (repo-wise) |
|---|---|---|---|---|
| Codex (Chen et al., 2021) | 58.73 | - | 60.64 | - |
| Oracle | 79.63 | 35.58 | 80.24 | 32.31 |
| Random | 58.13 | -1.02 | 58.95 | -2.79 |
| Random NN | 58.98 | 0.43 | 60.04 | -0.99 |
| File-level BM25 | 63.14 | 7.51 | 64.28 | 6.00 |
| Identifier Usage (Random) | 64.93 | 10.55 | 67.83 | 11.85 |
| Identifier Usage (NN) | 64.91 | 10.52 | 67.94 | 12.03 |
| Fixed Prompt Proposal | 65.78 | 12.00 | 68.01 | 12.15 |
| RLPG-BM25 | 66.41 | 13.07 | 68.15 | 12.39 |
| RLPG-H | **68.51** | **16.65** | 69.26 | 14.21 |
| RLPG-R | 67.80 | 15.44 | **69.28** | **14.26** |

ment in performance. The right part of Figure 2 shows the performance of an oracle constructed from our prompt proposals. We see that across all data splits, the prompt proposals contribute to significantly large improvements over Codex (upto 36% for test split). These results might seem surprising as Codex has not been trained on prompts that consist of context other than the default Codex context. What makes this result more surprising is that in most of the cases, the prompt consists of mashed up context without logical ordering that may not even look like a semantically meaningful chunk of code (e.g. list of string literals from a sibling file followed by the default Codex context or post lines placed before the default Codex context as opposed to after). These results might suggest that as long as the relevant context (in our case repo-level knowledge in the form of prompt proposals) is present in any form in the prompt, it can be quite effective.

**RQ2 - Performance of PPC:** Having seen promise in our prompt proposals, next we present the results of RLPG, which for each target hole predicts a single best prompt proposal. Table 1 presents the success rates along with the percentage of relative improvements for the test data. The second and third columns correspond to the averages across all holes in the test data. The last two columns correspond to the average success rate of individual repositories. The latter metric doesn't account for the size of the repository. As can be seen from the table, all the RLPG variants as well as the fixed prompt proposal improve the performance significantly over Codex. The random baselines are either worse or on par with Codex. Identifier usage is a good baseline but still performs worse than either the fixed prompt proposal or RLPG. The improved performance of RLPG-BM25 as compared to fixed prompt proposal shows the value of generating example-specific prompts using RLPG. However, both the learned variants of RLPG, i.e., RLPG-H and RLPG-R outperform the RLPG-BM25, highlighting the importance of learning PPC. See Appendix D.5 for performance of all methods on individual repositories. Note that even though we consider identifier usage as a separate baseline, one could consider it as one of the prompt proposal leading to further improved performance of RLPG.

Despite our efforts of avoiding overlap, since the training data for Codex is not exactly known, there might be a slight possibility that part of our Google Code data is part of the training data for Codex. Even if there were an overlap, we want to point out that since Codex has seen the default Codex context during training, it would be more beneficial to use the default Codex context in the prompt rather than the context from the prompt proposals or any other context from other baselines. Therefore, under this scenario, our evaluation would be more generous to the Codex baseline with results biased more in favour of the Codex baseline than other methods we have used.

**Variation with #attempts:** Imagine a scenario where we have a human-in-the-loop who has been given $k$ attempts to prompt the LLM and then can choose one of the $k$ hole predictions. We wanted to see how does the performance of our framework varies with #attempts under this setting. This corresponds to using $k$ prompts generated with top-$k$ prompt proposals (one prompt per proposal) and marking success if any of the $k$ prompts lead to success. The left part of Figure 3 shows the variation of SR over the validation data with the value of $k$. For RLPG, the top-$k$ prompt proposals were chosen based on the decreasing order of probabilities given by PPC. For the fixed prompt proposal, the top-$k$ prompt proposals were decided based on decreasing order of success rate of the individual prompt proposals on the validation dataset. From the figure, we notice that as we increase the value

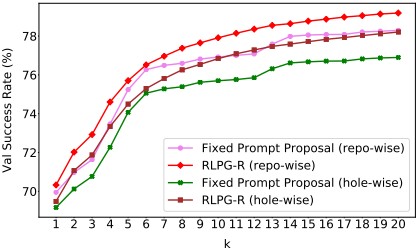 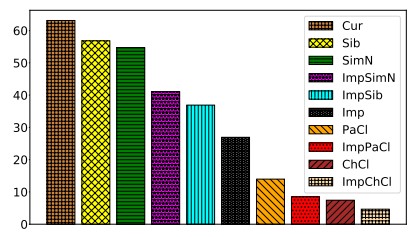

Figure 3: *(Left)* Variation of RLPG and Fixed Prompt Proposal with #attempts ($k$); *(Right)* Mean success rates of different prompt sources when they are applicable.

of $k$, the performance increases gradually at first and then saturates towards the oracle performance (79.05% for val data). This behaviour is observed for both fixed prompt proposal as well as RLPG. However, we see that for the same value of $k$, the success rate for RLPG is higher indicating that PPC learns a useful ranking of the prompt proposal contexts that can scale well with the #attempts.

**Performance based on Prompt Proposals:** The right part of Figure 3 shows mean success rate of prompt sources when we count success only when the corresponding prompt source is applicable. From the figure, we see that the current file is the most important prompt source. Closely following are sibling files and similar name files. We see that all prompt sources have non-zero chances of success, highlighting the usefulness of each prompt source. See Appendix D.1 for a similar breakdown based on prompt context type and Appendix E for analysis of successful and failed sample cases.

# 4  RELATED WORK

**LLMs for Code:** Recently, there has been a lot of work around large language models of code. One class of models are the decoder-only models that correspond to generating code from left-to-right. Codex (Chen et al., 2021), Google's model (Austin et al., 2021), GPT-J-6B (Wang & Komatsuzaki, 2021), GPT-Neo (Black et al., 2021b), GPT-Neo-X (Black et al., 2021a), CodeParrot (Tunstall et al., 2022), PolyCoder (Xu et al., 2022a) and InCoder (Fried et al., 2022) are some examples. We also have some encoder-only models that use a masked language modelling objective. CodeBERT (Feng et al., 2020), GraphcodeBERT (Guo et al., 2020) and CuBERT (Kanade et al., 2020) are examples of such models. Lastly, we have the class of encoder-decoder models that generally use a bidirectional encoding of a context to decode a series of masked tokens. Code-T5 (Wang et al., 2021) and AlphaCode (Li et al., 2022) are examples of such models.

**Repo-Level Info:** Fewer works use information from outside the current file. Hellendoorn & Devanbu (2017) propose a nested n-gram model that utilizes a locality-based cache where the locality consists of all directories from the root of the project (inclusive of the current file). Zhang et al. (2021) uses the parent class to generate the comments for the child class. Pashakhanloo et al. (2022b;a) capture the structure and semantics of the repository by converting it into a relational database and propose a graph-walk based mechanism for pruning the unrelated context. Lyu et al. (2021) incorporates the API-dependency graph in a LSTM-based Seq2Seq model to assist in code generation. Xu et al. (2022b) incorporate three types of structural locality features while training the kNN-LM (Khandelwal et al., 2020). These features are binary variables that correspond to the presence or absence of similar hierarchy. The three levels of hierarchy are (a) sibling file, (b) file in the same repo (c) no hierarchy. In contrast we have a much richer set of prompt proposals incorporating the semantics and structure of the repository. Also, we assume black-box access to the actual LM and restrict ourselves to generating a prompt for the LLM without performing any finetuning of the LLM.

**Prompt Generation:** There have been promising works around prompt generation techniques in NLP. Broadly, there are two categories of automatic prompt generation techniques. The first category corresponds to producing continuous/soft prompts where the prompt is described in the latent space of a language model (Li & Liang, 2021; Qin & Eisner, 2021; Bragg et al., 2021; Lester et al., 2021; Liu et al., 2021b). For example, Prefix-Tuning (Li & Liang, 2021) adds a prefix to the LM that can be learned by finetuning on examples from the downstream task. The second category produces discrete

prompts where the prompt is a text string that can be interpreted by a human (Shin et al., 2020; Gao et al., 2021; Schick & Schütze, 2021). For example, Autoprompt (Shin et al., 2020) generates prompt using a fixed template consisting of trigger tokens. The trigger tokens are shared across all inputs and determined by a gradient-guided search involving the LM. Our work falls in the category of discrete prompt generation techniques as we produce a prompt consisting of code tokens that can be easily interpreted by a human. However, in contrast to prior works that use a set of fixed templates for all examples, we learn to produce prompts conditioned on each example. Another important distinction is that we do not require access to the weights of the LM. A concurrent work as ours (Wang et al., 2022) studies the role of prompt-tuning when compared to fine-tuning for code translation, defect localization and code summarization. However, their technique requires access to the weights of the LLM and they perform experiments over models that are much smaller in scale than Codex. To the best of our knowledge, our work is the first to explore automatic prompt generation in a black-box access setting in the domain of source code.

## 5 Conclusions and Future Directions

We present RLPG, a framework that learns to automatically generate prompts conditioned on the example, without requiring access to the weights of the LLM. RLPG utilizes the structure of the repository as well as the context from other files in the repository using a set of easy to understand prompt proposals. Note that even though we have scoped and worded our prompt proposals to be repository-level, the idea of RLPG and prompt proposals in itself is quite universal and need not be scoped to a repository. Taking context from other repositories as well as external knowledge such as API dependencies offers an interesting direction to explore in the future. In this work, we are taking context from only one prompt proposal. For future work, we want to learn a model that can automatically compose a prompt from multiple prompt proposals (see Appendix D.3 for promising initial results). Other interesting directions include incorporating the user's feedback in RLPG and extending RLPG to multi-line code autocompletion.

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

## A  DATASET CREATION DETAILS

### A.1  CREATION OF HOLE COMPLETION DATA

To collect the hole completion data, we scraped Google Code [8] for repositories tagged with the language "Java". Then we deduplicated repositories by searching for a matching repository with the same name on GitHub. For those repositories with zero matching names on GitHub, we downloaded the archive and extracted the source code (preserving the directory structure). Next, we tried to determine the licenses of all repositories by either looking for a LICENSE file or matching with keywords "license", "copyright", "mit", etc. For repos for which our process was able to come up with a known license, we selected the ones having a permissive license, i.e., MIT, ApacheV2 and BSD. This was followed by removing files that are exact duplicates of each other within a repo. One of the reasons we found this inter-repository duplication may be because sometimes developers adopt lousy practises where instead of declaring a package and importing functions, they simply copy-paste the desired file in the current folder. The target holes coming from any of the duplicate files do not form part of the hole completion dataset. However, these files might be used to contribute to prompt proposal context for completing a target hole in a non-duplicate file. For the remaining files, we took each line that is not a blanked line or a comment, and chose the middle character as the hole position, i.e., all the characters from the middle of the line to the end of the line form target hole. To avoid large repos having strong bias on our prompt proposal classifier, we capped the contribution from each repo to be a maximum of 10000 holes. If the number of holes in the repo exceeds 10000, we randomly select 10000 holes.

### A.2  CREATION OF DATA FOR REPO-LEVEL PROMPT PROPOSALS

We used the tree-sitter API for Java [9] to get the parse-tree of an individual file in a repo. To get information at a repo-level, for each file in the repo, we stored the following information:

1. list of all class names in the file. This helped us to get the parent or child class file corresponding to a given parent or child class.
2. the file corresponding to each import statement.
3. for each import statement in the file, the position in the file where the import is used. This is used for ranking the files based on the heuristics mentioned in Table 2.
4. list of sibling files
5. list of similar name files. This was done by splitting the filenames based on either camel-case or underscore. If the sub-parts of two files match, then they are said to have similar name.

The above meta-data was calculated only once for each repo. The subsequent hole completions can use the same cached information. In practise, we can use a hash to store and retrieve this info efficiently. For a prompt proposal, given the prompt source, we first obtain a single file or ranked list of files (see Table 2) using the info in the parse tree in conjugation with the above repo-level meta-data. All the prompt proposal context type information (MN, MNB, SL, I, TI, FD) can then be obtained by querying the parse tree of the selected file.

## B  PROMPT PROPOSAL DETAILS

### B.1  RANKING OF FILES BASED ON PROMPT SOURCE

In Table 2, we provide details of how we select files for a given prompt source. Depending on the prompt proposal, we get either a single file or a list of files ranked based on some criteria. For example, if the prompt source is Import, we take all the import statements used in the current file and identify the location in the current file where the corresponding imports have been used. According to our heuristic, the closer is the import usage to the hole position, the more likely it is for the prompt proposal context coming from the corresponding import file to be more relevant (to predict the target hole). We get a ranked list of import files sorted based on increasing order of distance (i.e., number of lines ) between the import usage and the hole position. We start by taking all of the prompt proposal

---

[8]https://code.google.com/archive/
[9]https://github.com/tree-sitter/tree-sitter-java

context from the first file in the ranked list and then keep iterating the ranked list until either the total context length allocated to the prompt proposal gets exhausted or we reach the end of the ranked list.

Table 2: **Selecting files for a prompt source**

| Prompt Source | File Ranking |
|---|---|
| Current | file with the target hole. Returns a single file. |
| Parent Class | file that contains the parent class that occurs closest to the target hole. Returns a single file. |
| Import | files with the corresponding import usage ranked based on the proximity to the hole. Returns a ranked list of files. |
| Sibling | files with import usage common to the current file and the sibling file, ranked based on the proximity to the hole. The total number of common imports between the current and the sibling file is used as a tie-breaker. Returns a ranked list of files. |
| Similar Name | files with import usage common to the current file and the similar name file, ranked based on the proximity to the hole. The total number of common imports between the current and the similar name file is used as a tie-breaker. Returns a ranked list of files. |
| Child Class | files with import usage common to the current file and the child file, ranked based on the proximity to the hole. The total number of common imports between the current and the child class file is used as a tie-breaker. Returns a ranked list of files. |
| Import of Sibling | import files ranked based on the frequency of usage in all the sibling files. Returns a ranked list of files. |
| Import of Similar Name | import file ranked on the basis of frequency of usage in all the similar name files. Returns a ranked list of files. |
| Import of Parent Class | import file ranked on the basis of frequency of usage in all the parent class files. Returns a ranked list of files. |
| Import of Child Class | import file ranked on the basis of frequency of usage in all the child class files. Returns a ranked list of files. |

## B.2   EXAMPLES OF PROMPT CONTEXT TYPE

We provide examples of each of our prompt context type below:

1. Post Lines (PL) : For the example shown in Figure 1 of the main paper, post lines will take all the lines after the line `mg.InitializeToAssignment(CurrentAssignments())` till we reach the end of the file (`AffinityPropagation.java`).
2. Identifiers (I): Identifiers are the names of variables used in the code. For example, for the prompt proposal context taken from the imported file shown in Figure 1 in the main paper (highlighted in violet), identifiers are `InitializeToAssignment` (line 1), `a` (line 1), `currentAssignment_` (line 2), `a (` line 2), `clone` (line 2), `alreadyInitialized_` (line 3), `justOneRound_` (line 4).
3. Type Identifiers (TI): Type Identifiers define the type of an identifier. For example, in the code snippet `class DPAffinityPropagation extends AffinityPropagation`, `[ AffinityPropagation` is labeled as a type identifier. Similarly in the snippet `DPAPParameters parameters_;`, `DPAPParameters` is a type identifier.
4. Field Declarations (FD): The variables of a class type are introduced by field declarations. For example, `double[][] mHijMujT_;` and `MessageValuePair[][] sortedMHijMujTs;` are examples of field declarations.
5. String Literals (SL): A string literal is the sequence of characters enclosed in double-quotes. For example, in the code snippet, `System.err.println("DPAP load Warning:  unknown parameter " + entries[0] + ", value = " + entries[1]);`, we have two string literals: (a) `"DPAP load Warning:  unknown parameter "` ; (b) `", value = "` .

6. Method Names (MN): For the example shown in Figure 1 of the main paper, `public void InitializeToAssignment(int[] a)` is the method name prompt context type.
7. Method Names and Bodies (MNB): For the example shown in Figure 1 of the main paper, the part highlighted in violet represents the method names and bodies.

### B.3 TRUNCATION STRATEGIES FOR PROMPT PROPOSAL CONTEXT

If the prompt proposal context is greater than the context length allocated to it, then we need to truncate the prompt proposal context. We followed the below two schemes for truncating context:

- **front:** We truncate the context from the front. This is used for all prompt sources except Parent Class and when we take PL from Current.
- **back:** We truncate the context from the back. This is used when the prompt source is Parent Class and when we take prompt context types other than PL from Current.

The truncation strategies for each case were selected based on results on a small validation set. For the prompt source Current, except when the prompt context type is PL, we always start by taking code of prompt context type from after the hole position. This makes sense as the default Codex context will anyways contain code before the hole. Only if this turns out to be blank, we will use the code of context type from before the hole.

### B.4 LIST OF PROMPT PROPOSALS

Table 3: **List of our proposed repo-level prompt proposals**

| Prompt Proposal ID | Prompt Source | Prompt Context Type |
|---|---|---|
| $0, 1, 2, 3, 4$ | Current | MN, I, TI, SL, FD |
| $5, 6, 7$ | Current | PL (taking 25%, 50% and 75% contribution to the total context length) |
| $8, 9, 10, 11, 12, 13$ | Parent Class | MNB, MN, I, TI, SL, FD |
| $14, 15, 16, 17, 18, 19$ | Import | MNB, MN, I, TI, SL, FD |
| $20, 21, 22, 23, 24, 25$ | Sibling | MNB, MN, I, TI, SL, FD |
| $26, 27, 28, 29, 30, 31$ | Similar Name | MNB, MN, I, TI, SL, FD |
| $32, 33, 34, 35, 36, 37$ | Child Class | MNB, MN, I, TI, SL, FD |
| $38, 39, 40, 41, 42, 43$ | Import of Sibling | MNB, MN, I, TI, SL, FD |
| $44, 45, 46, 47, 48, 49$ | Import of Similar Name | MNB, MN, I, TI, SL, FD |
| $50, 51, 52, 53, 54, 55$ | Import of Parent Class | MNB, MN, I, TI, SL, FD |
| $56, 57, 58, 59, 60, 61$ | Import of Child Class | MNB, MN, I, TI, SL, FD |
| $62$ | Codex | - |

### B.5 OTHER PROMPT PROPOSAL VARIATIONS

We experimented with other variations that include: (a) appending class names at the beginning of the prompt proposal context, (b) using newline or space to join the prompt proposal context and the default Codex context, (c) taking all or the top-$k$ of the prompt context types, (d) ordering of top-$k$.

- **Context Separator:** This defines how we join the prompt proposal context string to the default Codex context string. We experimented with space and newline as context separators.
- **Prompt Proposal Context Formatting:** We can format the prompt proposal context before giving it to the Prompt Composer. We experimented with the following options:
  1. class_name: append [class name of the file] at the beginning of the prompt proposal context taken from each file that is part of the prompt source. For example, if we are taking prompt proposal context from two import files $f1$ and $f2$, the prompt proposal context will be formatted as: [class name of $f1$] prompt proposal context from $f1$ + space + [class name of $f2$] prompt proposal context from $f2$. We use this when the prompt proposal context types are MN, I, TI, FD and SL.
  2. class_method_name: we apply this only when the prompt proposal context type is MNB. We append method names at the beginning of each of the corresponding method bodies. We also

append the prompt proposal context from a file with the name of the class as described in the previous item.

3. comment: Adding in the prompt proposal context as a comment, i.e., formatting it as: /** prompt proposal context */. This wasn't found to be much useful.
4. none: passing the prompt proposal context as it is. We use this when the prompt proposal context type is PL.

- **Top-k Type:** For each of the prompt proposal context types, except PL, we experimented with taking the (a) first (b) last and (c) all of the prompt proposal context types, i.e., we can take first-10 identifiers. We found 'all' to be the best among all.
- **Top-k:** We experiment with k values of (a) 10 (b) 20 and (c) all. We found 'all' to work best for all prompt context types.

## C    IMPLEMENTATION DETAILS

### C.1    RLPG-H

We used Adam (Kingma & Ba, 2015) optimizer with a learning rate of 3e-4 and batch size of 64. We used CodeBERT (Feng et al., 2020) as our pretrained model $F_\phi$ to obtain the representation of hole window. The size of the representation (corresponding to the hidden dimension of the `[CLS]` token) is 768. $W^1 \in \mathbb{R}^{512 \times 768}, b^1 = 512, W^2 \in \mathbb{R}^{63 \times 512}, b^2 = 63$.

### C.2    RLPG-R

We used Adam (Kingma & Ba, 2015) optimizer with a learning rate of 3e-4 and batch size of 64. We used CodeBERT (Feng et al., 2020) as our pretrained model $F_\phi$ to obtain the representation of hole window and prompt proposal context. The size of the representation (corresponding to the hidden dimension of the `[CLS]` token) is 768. In equations 1, 2 and 3 in Section 3.2, the projection matrices $W_i^Q \in \mathbb{R}^{d_q \times d_{model}}, W_i^K \in \mathbb{R}^{d_k \times d_{model}}, W_i^V \in \mathbb{R}^{d_v \times d_{model}}, W^O \in \mathbb{R}^{d_{model} \times \tau d_v}$. For the multihead attention, we used $d_k = d_q = d_v = 32$, $\tau = 4$ and $d_{model} = 768$, $W_r \in \mathbb{R}^{63 \times 768}$ and $b_p = 63$. For each head, we perform a scaled dot-product attention (Equation 4). $G$ module consists of a dropout (Srivastava et al., 2014) layer, a residual connection (He et al., 2016), a layernorm (Ba et al., 2016), followed by a sequence of (a) dense layer of weights=$2048 \times 768$, bias=768, (b) relu activation, (c) dense layer of weights=$768 \times 2048$, bias=2048, (d) dropout layer, (e) residual connection, (f) layernorm. A dropout value of 0.25 was used while training. Our model resembles one layer of the transformer encoder block (Vaswani et al., 2017).

### C.3    BASELINES

Random baseline first selects a file randomly from the current repository followed by selecting a random line within that file. We choose all the lines starting from that line to the end line of the chosen file as context (excluding the hole window if the chosen file is the current file). The nearest neighbour similarity is based on the dot product between the representation of the hole window and the representation of the context, where we use a pretrained CodeBERT (Feng et al., 2020) model to obtain the representations. For the Identifier Usage baseline, if the nearest identifier to the hole doesn't return any usage window, we proceed to the next nearest identifier. For faster computation and to avoid memory issues when running on our hardware, for NN baselines, we collect 64 random neighbours and then rank based on the nearest neighbour distance. The BM25-based baselines use the Okapi BM25 implementation with default parameters given by the pip package `rank-bm25 0.2.2` [10]. For file-level BM25, if the file context exceeds the allocated context length, we truncate from the back.

---

[10]https://pypi.org/project/rank-bm25/

# D ADDITIONAL RESULTS

## D.1 ABLATION ON PERFORMANCE BASED ON PROMPT PROPOSAL

Figure 4 shows the mean success rate of prompt context types when success is counted only for the cases when these prompt contexts are applicable. As can be seen from the figure, post lines is the most useful prompt context type on an average. The contribution from other prompt context types though smaller than post lines is still significant highlighting the importance of each prompt context type.

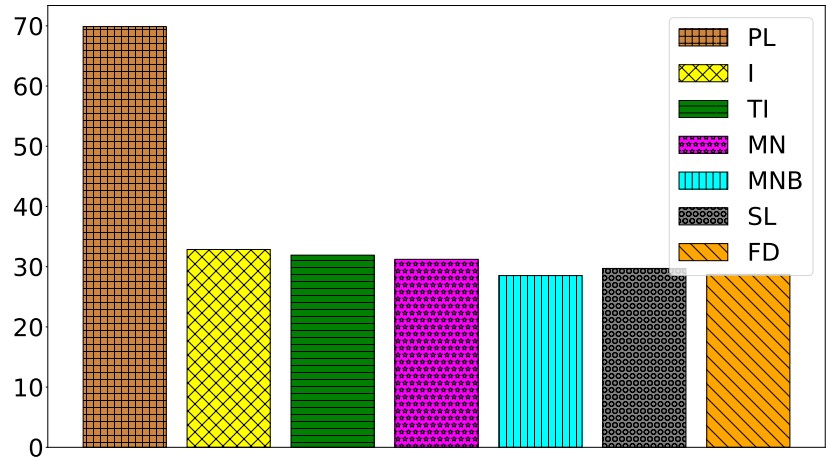

Figure 4: Mean success rate on validation data based on prompt context type when they are applicable.

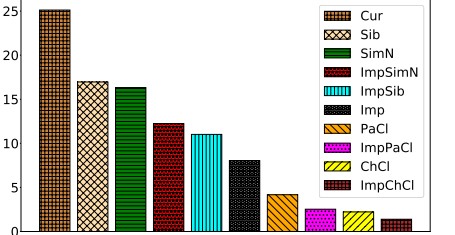 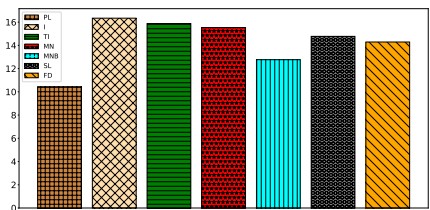

Figure 5: *(Left)* Normalized success rate of prompt sources when applicable, *(Right)* Normalized success rate of prompt context types when applicable

Figure 5 shows the normalized success rates where the normalization is performed across the prompt proposals. This helps us understand the relative performance of prompt proposal sources and context types. The left part of the figure breaks down the performance based on prompt sources and the right part breaks down based on prompt context types. One thing to note from the plot of prompt context types is that when we consider relative performance, post lines is no longer the most dominant context type. This is because post lines is tied to only when the prompt source corresponds to the current file, thereby contributing to lower numbers when compared to most of the other context types that are tied to all prompt sources.

## D.2 PERFORMANCE ON NON-IMMEDIATE POST LINES

Table 4 shows the performance of post lines when starting the fourth line after the target hole line (i.e., skipping three lines after the target hole) as opposed to starting from the line that immediately follows the target hole line. This experiment helps us understand the performance when we are interested in doing a much harder task of multi-line code autocompletion, wherein the objective is to predict not just the blanked out portion in the current line but also say the next three lines. This can correspond to completing a block of code like a function body. As can be seen from the table, when starting from the fourth line, we see a very slight deterioration in performance. This is expected because the farther away we move from the target hole, the less relevant the post lines context would be. However, the performance drop is not significant suggesting that post lines is still a very useful prompt context type that can be used under the setting of multi-line code-autocompletion. Equivalently, we can include this as one of the prompt proposals in our framework along with the current version of post lines.

Table 4: Success Rate (SR) when taking different versions of post lines.

| Method | Success Rate(%) (hole-wise) | Rel. ↑(%) (hole-wise) | Success Rate(%) (repo-wise) | Rel. ↑(%) (repo-wise) |
|---|---|---|---|---|
| Codex (Chen et al., 2021) | 58.73 | - | 60.64 | - |
| Post Lines (immediate line after the hole) | 65.78 | 12.00 | 68.01 | 12.15 |
| Post Lines (skipping three lines after the hole) | 65.11 | 10.86 | 66.42 | 9.53 |

## D.3 COMPOSITION OF PROMPT PROPOSALS

Table 5 shows the performance of the two versions of RLPG when we compose the prompt proposal context from $l$ prompt proposals. We take the top-$l$ prompt proposals given by RLPG based on decreasing order of probability. To decide how much context should be used for each prompt proposal, we divide the total context length in proportion to the normalized probabilities of the top-$l$ prompt proposals. As can be seen from the table, even though PPC is not explicitly trained to perform composition (both the ground-truth vector and the representation of prompt proposal context involve a single prompt proposal), all the compositions lead to significant improvements over Codex. However, as expected the best results correspond to taking context from a single prompt proposal (i.e., the training setting). The drop in success rate with $l = 2$ and $l = 5$ is not that significant, which suggests that explicitly training RLPG to learn to compose contexts from different prompt proposals can lead to promising results and hence offers an interesting future direction.

Table 5: Success Rate (SR) of different compositions of the prompt proposals on the test set.

| Method | Success Rate(%) (hole-wise) | Rel. ↑(%) (hole-wise) | Success Rate(%) (repo-wise) | Rel. ↑(%) (repo-wise) |
|---|---|---|---|---|
| Codex (Chen et al., 2021) | 58.73 | - | 60.64 | - |
| RLPG-H ($l = 1$) | 68.51 | 16.65 | 69.26 | 14.21 |
| RLPG-R ($l = 1$) | 67.80 | 15.44 | 69.28 | 14.26 |
| RLPG-H ($l = 2$) | 67.07 | 14.20 | 67.87 | 11.91 |
| RLPG-R ($l = 2$) | 66.57 | 13.35 | 67.88 | 11.94 |
| RLPG-H ($l = 5$) | 66.60 | 13.40 | 67.91 | 11.98 |
| RLPG-R ($l = 5$) | 65.78 | 12.01 | 67.69 | 11.62 |
| RLPG-H ($l = 10$) | 65.53 | 11.58 | 67.24 | 10.88 |
| RLPG-R ($l = 10$) | 63.59 | 8.27 | 65.98 | 8.79 |

## D.4 EFFECT OF CONTEXT LENGTH

To understand the effect of context length on the performance of our prompt proposals, we took half of the context length available for a prompt in Codex and observed the performance of the oracle and fixed prompt proposal. As before, we saw that an oracle constructed from our prompt proposals shows remarkable improvement over Codex highlighting the value of our prompt proposals. However, when compared to a larger context length, the relative gains are smaller. This is expected as a smaller

context length means that the relevant context coming from a prompt proposal needs to be truncated to make it fit inside the prompt, thereby leading to loss of information.

Table 6: Success Rate (SR) of Codex and oracle over the test set when the total context length = 2048.

| Method | Success Rate(%) (hole-wise) | Rel. ↑(%) (hole-wise) | Success Rate(%) (repo-wise) | Rel. ↑(%) (repo-wise) |
|---|---|---|---|---|
| Codex (Chen et al., 2021) | 57.77 | - | 58.90 | - |
| Oracle | 61.90 | 7.15 | 67.18 | 14.07 |

## D.5 PERFORMANCE ON INDIVIDUAL REPOSITORIES

Table 7: Success Rate of different methods on training data

| Repo name | #Total Holes | Oracle | Codex | Fixed prompt proposal | RLPG-H | RLPG-R |
|---|---|---|---|---|---|---|
| largemail | 1653 | 75.38 | 55.11 | 62.73 | 63.94 | 63.28 |
| ftpserverremoteadmin | 7323 | 86.44 | 66.11 | 76.09 | 76.21 | 76.76 |
| myt5lib | 838 | 91.65 | 53.58 | 61.34 | 73.51 | 74.46 |
| seamlets | 4890 | 92.74 | 62.25 | 62.72 | 71.55 | 74.27 |
| gloodb | 10000 | 91.07 | 57.50 | 57.50 | 70.32 | 72.31 |
| jjskit | 9043 | 80.36 | 65.61 | 72.18 | 72.00 | 72.44 |
| mobileexpensetracker | 2298 | 75.94 | 57.88 | 67.28 | 66.84 | 66.97 |
| gfsfa | 10000 | 80.55 | 57.33 | 57.33 | 59.28 | 65.24 |
| swe574-group3 | 2029 | 76.79 | 54.46 | 66.19 | 65.16 | 64.91 |
| strudem-sicsa | 6131 | 77.83 | 64.96 | 72.55 | 73.25 | 73.32 |
| soap-dtc | 1370 | 81.24 | 64.82 | 70.73 | 71.61 | 72.70 |
| openprocesslogger | 7191 | 81.06 | 62.19 | 71.77 | 72.22 | 72.62 |
| tapestry-sesame | 397 | 72.54 | 45.84 | 61.21 | 60.71 | 63.98 |
| exogdx | 735 | 84.76 | 63.81 | 75.51 | 75.92 | 76.60 |
| designpatternjavapedro | 1069 | 78.30 | 54.82 | 64.36 | 63.99 | 68.57 |
| quidsee | 3020 | 81.66 | 60.79 | 69.50 | 70.36 | 70.26 |
| healpix-rangeset | 4734 | 63.54 | 48.71 | 54.67 | 54.94 | 55.07 |
| sol-agent-platform | 10000 | 73.76 | 58.22 | 65.72 | 65.65 | 65.94 |
| rsbotownversion | 10000 | 75.23 | 57.89 | 65.58 | 66.22 | 66.31 |

Table 8: Success Rate of different methods on validation data

| Repo name | #Total Holes | Oracle | Codex | Fixed prompt proposal | RLPG-H | RLPG-R |
|---|---|---|---|---|---|---|
| tyrond | 721 | 83.91 | 60.33 | 71.15 | 71.57 | 72.68 |
| math-mech-eshop | 2225 | 83.46 | 62.20 | 72.76 | 73.53 | 73.17 |
| infinispan-storage-service | 373 | 82.31 | 71.85 | 78.55 | 76.94 | 77.75 |
| teammates-shakthi | 7665 | 82.02 | 63.74 | 72.38 | 72.47 | 72.46 |
| javasummerframework | 10000 | 79.27 | 55.92 | 65.30 | 65.74 | 65.55 |
| tinwiki | 10000 | 73.67 | 69.27 | 69.27 | 69.12 | 69.58 |
| jloogle | 3145 | 84.55 | 73.16 | 77.87 | 77.17 | 77.36 |
| jcontenedor | 5464 | 81.26 | 58.99 | 67.77 | 67.95 | 68.32 |
| sohocms | 772 | 76.68 | 57.90 | 67.10 | 67.49 | 67.62 |
| affinity_propagation_java | 1466 | 79.54 | 59.14 | 70.33 | 70.26 | 70.26 |
| jata4test | 1921 | 71.06 | 44.09 | 54.92 | 55.91 | 57.47 |
| swinagile | 2595 | 79.69 | 63.01 | 72.29 | 72.49 | 72.68 |
| navigablep2p | 1322 | 75.72 | 59.76 | 65.43 | 65.13 | 65.28 |
| springlime | 879 | 83.50 | 62.34 | 74.18 | 74.86 | 74.40 |

Table 9: Success Rate of different methods on test data

| Repo Name | #Total Holes | Oracle | Codex | Fixed PP | RLPG-H | RLPG-R | Random | Random NN | Iden Usage (Random) | Iden Usage (NN) | File-Level BM25 | RLPG-BM25 |
|---|---|---|---|---|---|---|---|---|---|---|---|---|
| dovetaildb | 10000 | 76.89 | 57.12 | 66.45 | 66.06 | 66.25 | 57.45 | 57.58 | 61.39 | 60.77 | 59.39 | 66.09 |
| project-pt-diaoc | 10000 | 82.01 | 52.67 | 52.81 | 65.08 | 61.25 | 51.58 | 52.93 | 55.54 | 56.21 | 57.04 | 58.29 |
| realtimegc | 2513 | 77.64 | 57.58 | 67.01 | 67.85 | 68.48 | 57.78 | 58.89 | 63.51 | 63.99 | 61.84 | 66.69 |
| fswuniceubtemplates | 2070 | 77.44 | 55.7 | 58.89 | 66.81 | 65.8 | 55.22 | 55.89 | 65.7 | 66.43 | 59.28 | 66.71 |
| qwikioffice-java | 1138 | 76.45 | 70.21 | 70.21 | 69.86 | 70.56 | 46.13 | 48.15 | 60.37 | 62.92 | 64.41 | 58.17 |
| glperaudsimon | 1766 | 78.65 | 53.57 | 62.51 | 62.4 | 61.66 | 55.66 | 57.76 | 69.42 | 68.4 | 69.14 | 61.55 |
| xiaonei-java-api | 839 | 73.42 | 57.57 | 62.1 | 62.69 | 63.29 | 57.09 | 57.21 | 71.28 | 72.35 | 63.77 | 63.29 |
| ircrpgbot | 6591 | 83.67 | 69.67 | 77.24 | 76.71 | 76.65 | 69.55 | 70.54 | 74.68 | 74.43 | 69.32 | 75.75 |
| robotsimulator2009w | 7514 | 75.63 | 56.28 | 67.55 | 67.53 | 67.55 | 56.4 | 56.18 | 64.61 | 64.71 | 62.96 | 66.12 |
| gwt-plugindetect | 73 | 84.93 | 60.27 | 68.49 | 65.75 | 68.49 | 58.9 | 57.53 | 63.01 | 63.01 | 50.68 | 75.34 |
| apiitfriends | 1385 | 85.05 | 65.05 | 74.8 | 75.67 | 75.31 | 65.7 | 68.59 | 70.25 | 70.11 | 66.93 | 73.57 |
| wicketbits | 754 | 83.02 | 59.81 | 72.94 | 72.81 | 73.08 | 60.21 | 61.94 | 81.96 | 79.31 | 84.48 | 73.47 |
| hucourses | 590 | 84.41 | 70.68 | 77.46 | 77.63 | 77.97 | 70 | 72.2 | 70.68 | 72.54 | 53.39 | 75.08 |
| xfuze | 3055 | 84.09 | 62.82 | 73.62 | 72.73 | 73.62 | 63.67 | 65.17 | 77.25 | 75.97 | 77.32 | 74.01 |

Table 7, Table 8 and Table 9 present the success rates of different methods over individual repositories in the training, validation and test splits, respectively. The repo-wise averages in Table 2 in the main paper were calculated by taking the average of numbers corresponding to each column. The hole-wise averages correspond to multiplying the repo-wise numbers of each method by the total holes in the repo to get the total number of successful holes by that method for that repo. We then add the total number of successful holes across repos and divide it by the total number of holes in the entire data split to get the hole-wise averages.

# E  ANALYSIS OF SAMPLE CASES

In Figure 1, RLPG selects the prompt proposal that corresponds to taking method names and bodies from the imported file (i.e. `MaximizingGibbsSampler.java`). Note that `mg.` before the hole position indicates that a method used in the imported file is likely to be invoked. In this case, the prompt proposal context (highlighted in violet) contains the method name `InitializeToAssignment` (part of target hole). This in conjunction with the default Codex context which contains the method `CurrentAssignments()` (part of target hole) leads to generation of a successful prompt. On the other hand, the prompt created from the default Codex context fails to predict the target hole in this case. In general, we observed that in the absence of a strong signal, Codex has a tendency to give preference to natural language comments occurring before the hole position, e.g. naming the method based on the comment. This in certain cases might hurt. We provide insatnces of positive and negative samples cases for RLPG below:

## E.1  POSITIVE CASES

We provide some examples of cases where RLPG led to the correct prediction and Codex failed.

1. Cases where part of the target hole is found exactly in the prompt proposal context.
   - RLPG = `Propagation(int numVars)` vs Codex = `Propagation()`
   - RLPG = `tersFromFile(String filename) {` vs Codex = `ters(String filename) {`
   - RLPG = `als("dampingFactor")) {` vs Codex = `als("numVars")) {`
   - RLPG = `] + ", value = " + entries[1]);` vs Codex = `]);`
   - RLPG = `stem.exit(1);` vs Codex = `stem.err.println("DPAP load error:  " + ex.get`

2. Cases where Codex takes strong hint from the preceding natural language comment, thereby producing incorrect predictions.
   - RLPG = `d PassMessages()` vs Codex = `d DoOneRoundOfMessagePassing()`
   - RLPG = `teger> CurrentExemplars() {` vs Codex = `teger> ChooseExemplars() {`

- RLPG = `ring FileName() {` vs Codex = `ring GetAlgorithmFilename() {`

## E.2 NEGATIVE CASES

In certain cases, extra information from prompt proposal-context might lead to confusion and produce incorrect predictions.

- RLPG = `an hasConverged_;` vs Codex = `an converged_;`
- RLPG = `_[i][j] = -Double.MAX_VALUE;` vs Codex = `_[i][j] = 0;`

