# OpenReview forum: "Repository-Level Prompt Generation for Large Language Models of Code"
_ICLR.cc/2023/Conference — Submitted to ICLR 2023_

### Official Review · Reviewer_9is8 · 2022-10-24

**Confidence:** 3
**Correctness:** 3
**Technical Novelty And Significance:** 3
**Empirical Novelty And Significance:** 2
**Recommendation:** 8

**Clarity, Quality, Novelty And Reproducibility:**

The paper is clearly written.
I think that the paper presents a novel approach for improving results of tasks such as code completion by generating prompts from whole repository using a trained framework.
The results of the paper should be reproducible assuming that the dataset used by authors to train the PPC is available.



**Strength And Weaknesses:**

Strengths:
- Addresses an important and novel area of good prompt generation for LLM tasks
- Approach does not require access to the weights of LLM
- Suggests and implements prompt generator that uses information from the whole repository. The prompt generator also uses repository level prompt "proposals" (rules/suggestions).
- Using generated prompts significantly improves Codex results

Weaknesses:

- It is not clear whether prompt proposals are really per-repository. It seems that they will be pretty universal and should not vary much. So the value of repository-specific prompt proposals is not really established or proven. This is a minor issue though.


**Summary Of The Paper:**

Authors propose prompt generator based on the whole code repository for the code completion tasks.
Authors show a significant improvement over baseline Codex results by using their prompt generator.

**Summary Of The Review:**

I think that the paper presents a novel approach for improving results of tasks such as code completion by generating prompts from whole repository using a trained framework. This is interesting work and should be accepted to ICLR.

---

> ### Author Response · Authors · 2022-11-15
> **Prompt proposals being really per-repository and availability of training data for PPC.**
>
>
> Thanks a lot for your positive feedback on our paper! We really appreciate the time and effort you spent reviewing our paper. We are really happy that you liked that we address an important and novel area of prompt generation for LLM, that our approach doesn’t require access to the weights of the LLM, we use information from the whole repository and our results show promise. We address some of your comments below:
>
> **Prompt-proposals being really per-repository**: This is a fair point! Even though we have scoped and worded our prompt proposals to be repository-level, the idea of RLPG and prompt proposals in itself is quite universal and need not be scoped to a repository. Taking context from other repositories as well as external knowledge in the form of API dependencies etc., would be an interesting future work. Thanks for mentioning this. We have included a clarification about this in the updated version of the paper.
>
> **Availability of training data for PPC**: Indeed we provide the data we used for training PPC as part of the supplementary material. Please see train folder in rule_classifier_data for the preprocessed data and gcode-data.tar.gz for raw scraped data. We also provide code for replicating our results with a detailed README.

---

> > ### Comment · Reviewer_9is8 · 2022-12-10
> > **Thank you for your responses**
> >
> > Thanks to authors for general and individual responses.

---

### Official Review · Reviewer_VAEL · 2022-10-24

**Confidence:** 3
**Correctness:** 3
**Technical Novelty And Significance:** 3
**Empirical Novelty And Significance:** 3
**Recommendation:** 6

**Clarity, Quality, Novelty And Reproducibility:**

The paper is generally well written. The implementation of the model is not available for replication study. The proposed work appears to be novel.

**Strength And Weaknesses:**

Strengths:

- A novel way of generating prompts.

- Extensive experiments.

- RLPG is efficient for both training and inference.

Weaknesses:

- The paper is better for a software engineering/programming language conference.

- Only the structure of the repository and the context from other relevant files are considered in the prompt proposals.


**Summary Of The Paper:**

This paper proposes a repository-level prompt generation (RLPG) approach for language models of source code. Without having access to the weight of LLMs, RLPG can improve the performance of LM in code completion task. Through training the prompt proposal classifier, the prompt composer can generate high quality prompts for LM, increasing the accuracy of code completion.

**Summary Of The Review:**

I appreciate that the proposed technique can generate prompts in a black box manner. Different from previous work about line-level code completion, this paper considers the completion from the cursor to the end of the line. It involves a wide range of prompt sources and prompt context types. The proposed approach is well ablated and extensive experiments are conducted. RLPG is also efficient for both training and inference.

The proposed work learns to generate example-specific prompts using prompt proposals, which are taken from the structure of the repository and the context from other relevant files (e.g., imports, parent class files). The authors may also consider other code information such as API. See, for example:

Lyu et al., Embedding API dependency graph for neural code generation. Empirical Software Engineering, 26(4):61, 2021.

For RLPG-H, the used code context is two lines before and after the hole. Are two lines (or four) enough? Maybe the target hole is related to some statements far away from the target (e.g. some global variable definition).

The Identifier Usage (Random) also achieves promising results, why?

Figure 1 is not easy to understand.

Recently, the following work also applies prompt learning to pre-trained code models: Wang et al., No More Fine-Tuning? An Experimental Evaluation of Prompt Tuning in Code Intelligence, https://arxiv.org/abs/2207.11680.

The proposed technique is more about software engineering/programming language, the novel technical contribution to neural networks/language models is a bit limited.

---

> ### Author Response · Authors · 2022-11-15
> **Part 1 - Using API dependency information, limited window sizes and answers to other questions.**
>
> We thank the reviewer for taking the time to provide constructive feedback on our paper! We are happy that the reviewer found our technique of generating prompts novel (especially the black-box access part), our experiments extensive and our framework efficient. We address your concerns below:
>
> **Using API dependency information**: Thanks a lot for the suggestion and the reference! We have cited the reference in our related works section. Please see the third point of Summary of Reviews and Responses (in part 2) above for more clarification and our proposal on how it can be integrated with RLPG. Even though in this work, we have restricted ourselves to repository-level context for prompt generation ( which in itself is novel, especially for black-box access to the LLM), as you rightly pointed out the idea of RLPG is quite general and need not necessarily be limited to the context within the repository and inclusion of external information such as API dependencies outside the repository when available, can indeed be quite useful.
>
> **Better for software engineering conference**: While our work has direct implications towards software developers, our proposed framework relies on a neural-network model (i,e, PPC) and we offer an approach for prompt generation with black-box access to an LLM. We believe that these points make our work a good candidate for the applications area of an ML conference like ICLR. Below, we provide references to other papers [1-12] that operate in a similar setting and have been published in ML conferences.
>
> [1] https://openreview.net/forum?id=ZVe_WeMold (Neurips 2022)
>
> [2] https://openreview.net/forum?id=Q8GnGqT-GTJ (Neurips 2022)
>
> [3] https://openreview.net/forum?id=Gsbnnc--bnw (Neurips 2022)
>
> [4] https://openreview.net/forum?id=e8PVEkSa4Fq (Neurips 2022)
>
> [5] https://openreview.net/forum?id=o4uFFg9_TpV (Neurips 2022)
>
> [6] https://openreview.net/forum?id=_VjQlMeSB_J (Neurips 2022)
>
> [7] https://openreview.net/forum?id=e2TBb5y0yFf (Neurips 2022)
>
> [8] https://openreview.net/forum?id=9Vrb9D0WI4 (ICLR 2022)
>
> [9] https://openreview.net/forum?id=P-pPW1nxf1r (ICLR 2022)
>
> [10] https://openreview.net/forum?id=ek9a0qIafW (ICLR 2022)
>
> [11] https://openreview.net/forum?id=DhzIU48OcZh (ICLR 2022)
>
> [12] https://proceedings.mlr.press/v162/xu22g/xu22g.pdf (ICML 2022)
>
> **Limited window sizes for RLPG-H**: We agree with the reviewer that just two lines before and after the hole (four lines in total) may not be enough for capturing the context around the hole. We made this design decision because we rely on CodeBERT for obtaining the context representations that can support a maximum context length of only 512 tokens. Using a source code model that can support a longer context length will likely improve the performance of RLPG-H as well as all RLPG-R even further.
>
> **Why Identifier Usage (random) is good?** The identifier usage baselines consist of two parts: (a) how we choose the candidates windows, and (b) how do we choose amongst the candidates. Our logic for selecting candidate windows considers usage windows around the closest identifier to the target hole. The chosen identifier usage windows can then be ranked either randomly (random) or based on the nearest neighbour distance (in the representation space of CodeBERT) of the identifier usage window to the hole window (NN). From our results, we see that (a) is more important than (b), i.e., it doesn’t make a significant difference in performance based on how we rank the identifier usage windows as long as we get all the relevant closest identifier usage windows. This makes sense as it is highly likely that the target hole might be present in one of these identifier usage windows taken from relevant parts of the repository. That being said, intuitively if we were to use a model that allows a bigger context length (than CodeBERT) to model the NN representations, it might be possible that identifier usage (NN) would show more improvement in performance over identifier usage (random), than what we currently observe.
>
>   We want to point out that rather than being included as a baseline, identifier usage can be easily incorporated within RLPG by considered it as one of the prompt proposals, leading to further improved performance of RLPG.
>
> **Implementation not available**: We have provided the complete details of our implementation including model implementation, data and a detailed guide in the README on how to replicate our results. This can be found in the supplementary material.
>
> **Figure 1 not easy to understand**: Can you please elaborate what parts of the figure are not easy to understand? Also, if you can suggest some changes that would help improve the clarity of the figure, that would be really helpful.

---

> > ### Author Response · Authors · 2022-11-15
> > **Part 2 - Concurrent related work**
> >
> > **Concurrent related work**: Thanks for pointing out this work! This paper studies the role of prompt-tuning when compared to fine-tuning for code translation, defect localization and code summarization. However, their technique requires access to the weights of the LLM and they perform experiments over models that are much smaller in scale than Codex. As per our understanding, this work came out on July 2022 ( two months after the recent related works period as mentioned in the ICLR guidelines https://iclr.cc/Conferences/2023/ReviewerGuide – see the answer to the last question). We have cited this as a concurrent related work and mentioned the differences in the updated version of our paper.

---

### Official Review · Reviewer_xrrm · 2022-10-26

**Confidence:** 4
**Correctness:** 2
**Technical Novelty And Significance:** 2
**Empirical Novelty And Significance:** 2
**Recommendation:** 3

**Clarity, Quality, Novelty And Reproducibility:**

Clarity:
It's not clear how they are selecting the prompt proposal at inference time. Are they taking the argmax over the probabilities for the different prompt proposals?

Quality:
Error bars are not given. It's unclear how much overlap the test set has with the pretraining data.


**Strength And Weaknesses:**

Strengths:

(1) The idea of generating training labels for the correct contexts and training a classifier to predict what the correct contexts should be is novel and interesting

(2) They investigate multiple possible baselines

Weaknesses:

(1) Overlap with pretraining data -- Github is not the only source of training data for Codex (https://openai.com/blog/openai-codex/ -- "OpenAI Codex is a descendant of GPT-3; its training data contains both natural language and billions of lines of source code from publicly available sources, including code in public GitHub repositories"). Thus it is very possible that Google Code -- which they use for their test data -- is part of the training data for Codex. Furthermore, deduping based on just repo name match is highly imperfect. A much better way would be file level deduping or suffix array based deduping. Unless there's a better understanding of what the pretraining data consists of, the results might be invalid. The authors could consider using a model like CodeGen for which the pretraining dataset is known. Alternatively they could use code published after June 2021 (the training data cutoff for the Codex davinci-002 model), dedupe it against the code data available upto June 2021 and use that as their test data.

(2) The improvements over both right context as well as identifier usage baselines are modest.

(3) They don't explore how different prompt proposals could be combined with each other

**Summary Of The Paper:**

The paper is part of the stream of papers that try and generate code given a prompt. This paper tries to assess that if additional context is given as part of the prompt, whether the performance can improve. In particular, they predefine different types of contexts and then train a classifier to predict which context should be prepended to the prompt. To get the training data, they given different types of contexts to Codex and if a particular context is able to make the completion match the ground truth completion, they label that context as 1. They then train a multi-label classifier and use its predictions at inference time to decide what context to add on to the prompt.

They compare against several baselines - no additional context, oracle context (using the ground truth correct contexts) which is the upper bound, random context, nearest neighbors in representation space from the random contexts, lines using the closest identifier throughout the repo, and context to the right of the line to be completed.

The strongest baseline is the right context baseline against which they show a 1-4% relative improvement and a 14-16% relative improvement over no additional context (Table 1)

**Summary Of The Review:**

While I believe the ideas in the paper are interesting and novel, the fact that we do not know the pretraining data which could easily overlap with the test data + the deduping with respect to the the known part of pretraining data being imperfect (based on repo name match instead of file content match or suffix array deduping), the results are not reliable.

---

> ### Author Response · Authors · 2022-11-15
> **Part 1 - Overlap with pretraining data**
>
> We thank the reviewer for the time they took to provide feedback on our paper! We are glad that you found the ideas in our paper interesting and novel and liked the fact that we investigate multiple baselines. We address your concerns below (in two parts):
>
> **Overlap with pretraining data**: We agree with the reviewer’s suggestion of using file-level or suffix-array based deduping as better alternatives, but since the exact training data used by Codex is not publicly available, it was not possible for us to use these techniques. We want to apologize if our wording in the paper suggested that we were sure that there is no overlap with the pretraining data. We have included the following sentence in the paper to make it more clear: “Despite our efforts of avoiding overlap, since the training data for Codex is not exactly known, there might be a possibility that part of our Google Code data is part of the training data for Codex.“
>
> We wanted to point out that **even if there were an overlap, since Codex has seen the default Codex context during training, it would likely be more beneficial to use the default Codex context in the prompt rather than the context from the prompt proposals or any other context from other baselines. Therefore, under this scenario, we believe our evaluation would be more generous to the Codex baseline** with results biased more in favour of the Codex baseline than other methods we have used. Even under this potential bias, we have shown that our methods and other baselines outperform Codex. We have included this clarification while discussing the results in the updated version of the paper.
>
> The reason we went with Codex is that Codex is the state-of-the-art model for code autocompletion. There are several papers based on Codex ([4-7] to cite a few), a vast number of applications that are now built on top of Codex [8], and it is quite popular via GitHub Copilot. As mentioned in our paper, we wanted to minimize any potential overlap with the training data and hence decided to go with Google Code archives data instead of Github and made additional efforts of deduplication as mentioned in Appendix A.1. Our decision was influenced by some information that we describe next. As mentioned in Section 3.2 of our paper, we have used code-davinci-001 engine. This version of the engine roughly corresponds to the publication of the Codex paper by OpenAI[1] ---”This paper describes several early Codex models”. Citing from Section 3.1 of the Codex paper [1]: “Our training dataset was collected in May 2020 from 54 million public software repositories hosted on GitHub, containing 179 GB of unique Python files under 1 MB.” As they have clearly stated, their training data consists of only Python repositories, while in our work we are using Java. As per the reviewer’s concern about overlap with the pretraining data, we would like to point out that as mentioned in Section 3.2 of the Codex paper [1], Codex was initialized with GPT-3. Based on Table 2.2 of [2], GPT-3 was not explicitly trained on non-natural language data sources. However, we agree with the reviewer that there might be some source code included in the Common Crawl data.
>
> [1] Chen, Mark, Jerry Tworek, Heewoo Jun, Qiming Yuan, Henrique Ponde de Oliveira Pinto, Jared Kaplan, Harri Edwards et al. "Evaluating large language models trained on code." arXiv preprint arXiv:2107.03374 (2021).
>
> [2] Brown, Tom, Benjamin Mann, Nick Ryder, Melanie Subbiah, Jared D. Kaplan, Prafulla Dhariwal, Arvind Neelakantan et al. "Language models are few-shot learners." Advances in neural information processing systems 33 (2020): 1877-1901.
>
> [3] https://openai.com/blog/openai-codex/
>
> [4] Julian Aron Prenner, Hlib Babii, and Romain Robbes. 2022. Can OpenAI's codex fix bugs? an evaluation on QuixBugs. In Proceedings of the Third International Workshop on Automated Program Repair (APR '22). Association for Computing Machinery, New York, NY, USA, 69–75. https://doi.org/10.1145/3524459.3527351.
>
> [5] Drori, Iddo, Sarah Zhang, Reece Shuttleworth, Leonard Tang, Albert Lu, Elizabeth Ke, Kevin Liu et al. "A neural network solves, explains, and generates university math problems by program synthesis and few-shot learning at human level." Proceedings of the National Academy of Sciences 119, no. 32 (2022): e2123433119.
>
> [6] Sarsa, Sami, Paul Denny, Arto Hellas, and Juho Leinonen. "Automatic Generation of Programming Exercises and Code Explanations Using Large Language Models." In Proceedings of the 2022 ACM Conference on International Computing Education Research-Volume 1, pp. 27-43. 2022.
>
> [7] Jain, Naman, Skanda Vaidyanath, Arun Iyer, Nagarajan Natarajan, Suresh Parthasarathy, Sriram Rajamani, and Rahul Sharma. "Jigsaw: Large language models meet program synthesis." In Proceedings of the 44th International Conference on Software Engineering, pp. 1219-1231. 2022.
>
> [8] https://openai.com/blog/codex-apps/

---

> > ### Author Response · Authors · 2022-11-15
> > **Part 2 - Combination of prompt proposals, identifier usage baseline and more**
> >
> > **Combination of prompt proposals**: Please see the second point of Summary of Reviews and Responses (in part 1) above.
> >
> > **Good results of identifier usage baseline**: We want to point out that considering usage windows around the closest identifier to the target hole is a very strong baseline, as it is highly likely that the target hole might be present in one of these usage windows taken from relevant parts of the repository. Infact, this itself can be considered to be one of the prompt proposals and incorporated within RLPG.
> >
> > **Prompt proposal selection at inference time**: Indeed, during inference we are taking the argmax of the probabilities over the prompt proposals. Thanks for pointing it out, we have clarified this in the paper.

---

### Official Review · Reviewer_oQyE · 2022-10-27

**Confidence:** 4
**Correctness:** 4
**Technical Novelty And Significance:** 3
**Empirical Novelty And Significance:** 3
**Recommendation:** 5

**Clarity, Quality, Novelty And Reproducibility:**

This paper is well-writtern and is of good quality. The proposed method is somewhat novel. And I expect that the results should be easy to reproduce.


**Strength And Weaknesses:**

Strength:
- The paper is well-written. The method is well explained with examples. Experiment results are extensive with analysis.
- The paper explores several variants on the prompt sources and prompt types, and made a nice figure to demonstrate which prompt source works the best, though only one is used in experiment at a time.


Weakness:
- Taking the whole file from the whole import file seems unnecessarily large context. Rather, if the import statement only imports one class or one function from the module, it makes sense to only consider the function implementation, rather than the whole file where the function resides. This especially applies to 7) Method Names and Bodies prompt type.
- The paper only considers one prompt proposal, which is a large limitation of the paper. Why don't the authors take top-k prompt proposals and put into context instead of just one? Besides, I'm curious whether there is a better choice of prompt proposal, for example, intuitively the function names, docstrings, and variable names of the imported function could help inform the model the functionality of the imported function.
- It seems that, RLPG-H is more like classification, while RLPG-R is more like retrieval. Would a lexical search method like BM25 could retrieve relevant prompts already compared? That way no training for the PPC is needed. And I would expect it works reasonably well.


**Summary Of The Paper:**

This paper proposes a method to make prompt proposals from repository-level context and add concatenate the prompt proposals with normal context to LLM in order to feed more related information to achieve better generations. The paper explores different sources of repository level prompts, as well as prompt context types. Experiment results show that additional repository level prompt help improve the performance significantly.


**Summary Of The Review:**

Overall, I think this is a decent paper with novel method proposed and well-performed experiments. I have some concerns on the paper, and would want to see more experiments and results to make the paper more complete.

---

> ### Author Response · Authors · 2022-11-15
> **Results on comparison with BM25, multiple prompt proposals, suggestions for additional prompt context types and more.**
>
> We really appreciate the reviewer’s comments and useful suggestions! We are glad that you found our paper well-written, our method novel and our experiments extensive. We address your comments below:
>
> **Comparison with BM25**: Please see the first point of Summary of Reviews and Responses (in part 1) above.
>
> **Multiple prompt proposals**: Please see the second point of Summary of Reviews and Responses (in part 1) above.
>
> **Suggestions for additional prompt context types**: Please see the third point of Summary of Reviews and Responses (in part 2) above.
>
> **Taking context from the whole file**: In case we were not able to clarify it properly in the paper, we wanted to bring to notice that we do not consider the context from the full import file. A prompt proposal consists of a prompt context source and a prompt context type. So if prompt context source = import file and prompt context type = method names and bodies then we will take only method names and bodies from the desired import file. Now instead of considering all method names and bodies from the import file, we can select a few as the reviewer suggested. We tried this out initially where instead of taking all method names and bodies, we take only first k or last k method names and bodies, where the order is determined by their occurrence in the file and k is either 10 or 20. We found that taking all context types works the best. See Appendix B.5 for more details.
>
> Thanks again for your comments, especially the suggestion for using BM25 that we think really helps the reader to gain further insights into the working of our framework. We would make sure to acknowledge the anonymous reviewer for their contribution to our work.

---

### Author Response · Authors · 2022-11-15
**Summary of Reviews and our Responses - Part 1**

We thank all the reviewers for the time they took to review our work and provide constructive feedback. We are glad that the reviewers found our paper well-written and of good quality (reviewer oQyE), our experiment results extensive with analysis (reviewer oQyE, VAEL, xrrm), our strong improvements over Codex (reviewer 9is8) and our approach of generating prompts as novel and interesting (reviewer xrrm, VAEL, 9is8). Reviewer 9is8 and VAEL liked that our “Approach does not require access to the weights of LLM”. Reviewer 9is8 commented that our paper “Suggests and implements prompt generator that uses information from the whole repository” and  “Addresses an important and novel area of good prompt generation for LLM tasks”. Reviewer VAEL also commented that “RLPG is efficient for both training and inference.” The main points raised by the reviewers were around comparison with a retrieval method like BM25, using context from multiple prompt proposals and suggestions for potential prompt sources and prompt context types. We address these below:

**Comparison with BM25** (Reviewer oQyE):  Thanks a lot for the suggestion! We added the following two methods based on BM25:

  - *RLPG-BM25*: Instead of using our prompt proposal classifier to rank prompt proposals, using the scores obtained by BM25 to select the best prompt proposal. The scores are calculated with the hole window being the query and prompt proposal contexts being the search documents. This serves as a non-learned retrieval method that makes use of our prompt proposals.
  - *File-level BM25*:  Same as above, except that instead of using our prompt proposal contexts, search documents consist of full context from other files in the repository. This baseline helps us understand the role of BM25 as a retrieval method independent of our prompt proposals.

    We have updated the paper to reflect the changes. Please see Section 3.3 of the paper for the results and discussion and Appendix C.3 for the details of implementation. We see that File-level BM25 outperforms Codex and the random baselines. This suggests the value of using a lexical-search based retrieval technique like BM25. Further, we see that when we use BM25 to rank our prompt proposals, the results are quite impressive. In fact, it outperforms the fixed prompt proposal baseline. However, as can be seen from the Table, both RLPG-H and RLPG-R variants are better than RLPG-BM25 which suggests the importance of learning PPC. We would like to thank the reviewer for their suggestion. We believe that the inclusion of methods will help the readers understand our framework better.


**Context from multiple prompt proposals** (Reviewer oQyE and Reviewer xrrm): We completely agree with the reviewers that this direction is interesting. In our future works section (last four lines), one of the potential next steps is to compose contexts from multiple prompt proposals. In fact, we carried out some experiments that we had included in Appendix D.3.  For these experiments, during inference, rather than taking the top-1 prompt proposal we take the top-$l$ prompt proposals given by RLPG, based on decreasing order of probability. Because of a lack of a clear signal on how much context length should be taken from these prompt proposals, we divide the total context length in proportion to the normalized probabilities of the top-$l$ prompt proposals. As can be seen in Table 5, even though PPC is not explicitly trained to perform composition (both the ground-truth vector and the representation of prompt proposal context involve a single prompt proposal), all the compositions lead to significant improvements over Codex. However, the best results correspond to taking context from a single prompt proposal (i.e., the training setting).
Because this simple first approach did not yield improvements, we believe that enough additional research is needed to make the idea work well that it's best left for future work. One possible direction would be to explicitly train RLPG to learn to compose contexts from multiple prompt proposals.

 We want to reiterate that our main contribution lies in proposing RLPG: a framework that can be used to incorporate context and structure from the repository without requiring access to the weights of the LLM. We started with the simplest way of including a single prompt proposal into the prompt and showed that it works really well. Including contexts from multiple prompts is indeed interesting and a promising next step.

---

> ### Author Response · Authors · 2022-11-15
> **Summary of Reviews and our Responses - Part 2**
>
> **Suggestions for additional prompt proposals** (Reviewer oQyE and Reviewer VAEL):
> We thank the reviewers for their useful suggestions! Each of the suggestions is quite interesting and some are already incorporated into our existing set of prompt proposals. We comment on the suggestions below:
>
>   - *Function names and class names*: As mentioned in Appendix B.5, we did try appending class names as well as class and method names while formatting the prompt proposal context. As the reviewer suggested, including this information along with the prompt context type leads to better results.
>
>   - *Docstrings*: When the prompt context type is method names and bodies, then we do consider the docstring that is part of the method as the context. Similarly, when we use post lines as a prompt context type, docstrings (or comments) would be included.
>
>   - *Variable names within an imported function*: Our existing prompt context types of identifiers, type identifiers, field declaration and string literals will always include variable names within the imported function. Similar in spirit to what the reviewer suggested, as mentioned in Appendix B.5, we did try to use a selected set of these prompt context types but found that using all of these context types was better than using a selected version.
>
>   - *Dependency between API methods*: This is indeed a valuable source of information. The API dependencies that are outside of the current file and part of the repository are in a way already captured via import statements. However, we can include the API dependencies within the current file as an additional source of information. One simple way of doing this would be to simply concatenate the context from the nodes in the API dependency graph (corresponding to the method names and bodies) obtained by traversing the graph starting from the method that is most proximal to the target hole. The API dependencies external to the repository (like JavaDocs) can also be quite useful. We want to point out that the idea of RLPG is quite general and need not necessarily be limited to the context within the repository. Therefore, RLPG can be easily extended to incorporate the API dependency graph (when available) as an additional prompt proposal. Thanks for the suggestion and the reference (we have cited it in our updated version).
>
>   - *Considering only the imported part as context*: Instead of considering all method names and bodies from the import file, we can select a few as the reviewer suggested. We tried this out initially where instead of taking all method names and bodies, we take only first k or last k method names and bodies, where the order is determined by their occurrence in the file and k is either 10 or 20. We found that taking all context types works the best. See Appendix B.5 for more details. As the reviewer suggested if we have the information on the exact class or function in the module that was used by the import file, using that information would be certainly useful. Our initial implementation was focused on defining prompt context types that are more general across all prompt sources like similar name files, sibling files, etc. where this information is difficult to get. Even within import files, it is quite common to import the full file at the top and then invoke a certain class or function within the module later in the code (which might itself be part of the target hole and hence not always available). However, when available it definitely makes a lot of sense to use context only from the invoked class or function. Thanks for the suggestion!
>
>   The prompt proposals that we have come up with in this work are meant to be recommendations, and in no way meant to be a completely exhaustive list of possible prompt proposals. Since RLPG allows the flexibility to accommodate additional prompt proposals, exploring other spaces of prompt proposals like these would be interesting future work.
>
> In addition to the above, we have responded in detail to specific comments raised by the reviewers in the reviewer-specific responses. We are very happy to respond to additional questions/comments during the discussion period.

---

### Decision · Program_Chairs · 2023-01-20

**Decision:**

Reject

**Justification For Why Not Higher Score:**

Experiments might have some issues.

**Justification For Why Not Lower Score:**

NA

**Metareview: Summary, Strengths And Weaknesses:**

The prompt source and context are important contribution that was found by corpus analysis. The overall idea is also interesting that could inspire future research in prompting code LLMs. However, after reading the paper carefully, some of my concerns share with those of reviewers oQyE and xrrm, in particular the dataset and using only one prompt proposal.

**Summary Of Ac-Reviewer Meeting:**

NA